# Sustainable Development of Urban Green Areas for Quality of Life Improvement—Argument for Increased Citizen Participation

**Mihaela Constantinescu ***[ID]**, Andreea Orîndaru, Ștefan-Claudiu Căescu and Andreea Pachițanu**

Marketing Department, The Bucharest University of Economic Studies, Bucharest 010404, Romania
* Correspondence: mihaela.co@gmail.com; Tel.: +40-721-223-896

**Abstract:** Considering the imperative need for sustainable urban development, this article argues for increased citizen participation in the decision-making process, as it generates better outcomes (due to a wider range of perspectives) and also makes people better citizens, as they will be partially responsible for the results. One major dimension of urban areas' which needs a sustainable development is represented by parks, which can be directly associated to citizens' quality of life (QoL). Thus, we have conducted direct research (face-to-face interviews) of park visitors in order to analyze the perceived impact of green areas on their quality of life. From all the QoL dimensions, we have selected six which are directly linked to park visits—health (mental and physical), social interaction, education and culture, family life, freedom, and connection with nature—in order to determine the perceived degree of association between them, as well as the specific activities done in the park that impact those six dimensions. The research results were used to develop a conceptual model which links quality of life to park visits, a model that can and should be used by public authorities in order to build a collaborative process for urban sustainable development.

**Keywords:** quality of life; urban green areas; urban sustainability; citizen participation; quantitative marketing research

---

## 1. Introduction

Urban sustainable development (as well as social sustainable development [1]) has been a hot topic in recent years since more and more people are concerned with their quality of life within the crowded urban setting [2]. In this urban setting, one particular aspect raises increased awareness and willingness to bring about transformation, and that is the issue of parks and green recreational areas, so needed in the era of increased levels of pollution and cities with a constantly growing number of inhabitants. But what does sustainable development mean when it comes to parks? And what does citizens' participation imply in transforming parks across cities? For answering these questions, science and citizens' feedback collected through research based on interdisciplinary and transdisciplinary transformation can lead to more realistic and actionable results [3]. While interdisciplinary studies involve researchers in different disciplines accepting concerted action and integration in order to achieve a shared goal related to a common study subject, transdisciplinary contributions not only combine knowledge and concepts used by academics and researchers, but also by other stakeholders from the civic society [4]. Both approaches might bring about a consistent contribution that will lead the way on the path towards creativity in urban sustainable development. Taking this last argument into consideration, this paper aims at bringing science one step further on the pathway towards urban sustainable development.

## 2. The Need for Sustainable Urban Transformation

People nowadays are progressively choosing urban areas as their home, which has led to an increasing urban population (for example, in Romania according to the National Institute of Statistics, the migration rate from rural to urban setting has been steadily growing between 2011 and 2018, hitting a level of 7.2 per 1000 inhabitants, very close to the 7.5 level of 2010). Crowded neighborhoods have become a common feature of all major cities of the world, raising now more than ever, issues of adopting a long-term perspective in managing and using all types of resources that the city has to offer to its inhabitants. Therefore, as Haaland and Konijnendijk van den Bosch [2] point out, as people are moving towards urban settings that lead to important environmental challenges, our attention and interest in the direction of sustainable development is raised now more than ever. Focusing on sustainable development within the urban areas is especially justified by its main aim: to bring more happiness in people's life through major improvements in their quality of life in perspective, being careful about other beings in the urban setting [5]. Basically, caring for all the beings in the urban setting is justified by their belonging, along with humans, to the entire urban ecosystem [6]. In the current research framework, urban sustainable development is defined as "positive quantitative, qualitative, directed, irreversible changes in the process of supply-production-distribution-consumption, the ones that help to adapt to the effects of endogenous and exogenous factors, to ensure higher rates of reproduction of resources relatively to the rate of consumption and, at the same time, to eliminate the disturbance of socio-ecological and economic security regarded as a result of the balance between consumption and reproduction of resources" [7].

Given the wider perspective, developing sustainably implies being free and creative about how we want to experience the future ahead [1]. In a different approach, developing sustainably also implies striving to achieve the ideal setting of humanism and complete harmony between mankind and nature [5]. Even though it might seem uncommon to link nature and cities in concepts relating to sustainability, recent technological development has created the premises of approaching sustainable development not only from the environmental perspective, but also from the perspective of quality of life and long term growth for the population [6], an idea also argued by Haaland and Konijnendijk van den Bosch [2] who stated that a real sustainable development approach means to achieve a two-fold objective: efficient resource creation and pleasant, welcoming design setting development for the city, aiming to achieve greater levels of quality of life.

Sustainable development of urban areas is even more needed in practical terms since for the long-term survival in the urban setting in this form, taking the sustainable development concept out of the abstract concepts and putting it to use into day-to-day practical approaches is a massive requirement [1]. Therefore, in order to achieve sustainability objectives, city managers and stakeholders must focus on the need to build a complete trust-based ecosystem between locals and the environment [6]. As Lepage [8] points out, sustainability relates also to the individual level, as it relates to people's ability to keep on growing and achieving their goals in terms of quality of life with the given resources and paying attention to those around them.

## 3. Citizens' Participation in Decision-Making Process for Urban Sustainable Development

Given the urban setting where consumption, production and human capital reach impressive levels of dynamism and growth, there is an urgent need for integrating efforts towards sustainability inviting a more holistic use of all the tools available nowadays [6]. In order to achieve that integrated and holistic approach, an ideal solution would need to combine authentic, people-oriented public policies and opening the door to citizens' involvement in the decision-making process, as Wagner and Andreas [3] emphasized.

In terms of civil society's commitment and participation, Huang and Fenney [9] argued that citizens' involvement in decisions related to policies which target them directly is likely to win more of their trust in government and to increase their willingness to comply with those decisions. Moreover, both researchers and practitioners have argued for one way to approach the decreasing trust in political

parties (on a growing trend these days): prepare the governmental institutions for increasingly inviting and involving citizens in relevant political decisions as a way to answer people's wish to participate more in such decisions [10].

But what does citizens' participation really involve? According to OECD, getting involved in government decisions has multiple levels in different stages of the policy cycle. Therefore, citizens' participation can become a reality in one of the following forms [11]:

1. Governments and their representatives make available all the information needed for the citizen to get informed about the latest policies. This form implies only a passive involvement of the citizen who just acknowledges the changes and whatever is important to them, but has no chance to switch to an active role in the decision-making process;

2. Beyond just being informed, citizens' have the opportunity to share with their representatives their feedback on different policies and decisions;

3. Genuine citizens' involvement implies creating the setting for collaboration and partnership with the government and its representatives, a setting where citizens practically engage in the decision- and policy-making process.

Out of these three options, the last one may lead to better results for both government and community. Irvin and Stansbury [12] confirmed this assumption, in discovering that citizen participation is often encouraged based on the belief that an engaged citizenry is better than a passive one.

Moreover, engaging citizens at local level is more likely to generate better results as opposed to national level citizen' engagement, especially since the local level grants the opportunity to create higher levels of transparency [13]. Going even further, the area where progress is needed more is the budget-setting area since citizens' involvement should relate more to expenditures and priorities in how their taxes should be used [14]. This approach proves to be even more suitable since inhabitants' involvement in decisions that have a great impact on their own life can substantially improve their life [13] so their community outcome is a reflection of their own desires. So, in the end, citizen participation influences the quality of service, strengthens the skills to use the service and ensures that the public needs are met [14]. Among other positive outcomes of citizens' participation, Van Eijk [13] mentions balancing diverse interests of different actors, achieving decisions' acceptance faster and developing more legitimate policies, along with creating a space for people to connect to each other. Irvin and Stansbury [12] made an even more detailed inventory of the citizens' participation advantages including education (for both the citizens and government representatives), political suasion, empowerment, breaking gridlock, avoiding litigation costs, and environmental management with better policy and implementation decisions. Van Eijk [13] argues that citizen participation generates two desirable outcomes: firstly, it is a learning experience for the citizens who in the end become better ones, and secondly, the decisions made with citizens' participation are better since they imply a wider range of perspectives. Still, the disadvantages for citizen participation also need to be considered and they include: time consuming, costly, the difficulty of diffusing citizen goodwill, complacency, poor representation, lack of authority, the power of wrong decisions and persistent selfishness [12]. When it comes to practical experiments with citizens' participation, Godbey and Kraus [15] point out that citizens' involvement in the decision-making process has been rather shallow, a conclusion also confirmed in the southeastern European countries' case where there is insufficient cooperation among politicians, citizens and local administration to establish a legal framework for achieving higher levels of active participation in the policy-making process [14].

The positive outcomes of citizens' participation can still bring a higher level of satisfaction within the community as long as the disadvantages are tackled with responsibility and a set of prerequisites are well established. In this respect, a series of ideal conditions were drafted by Irvin and Stansbury [12] in order for the citizen participation in the decision-making process to bring about the best results: citizen's willingness to volunteer, non-geographically dispersed stakeholders, citizen's ability to finance their meeting attendance, homogeneity of the community, knowledge that is easily owned, urgency of

the gridlock issue, high hostility towards the government, strong influencers in the community that are willing to be community's representatives, high credibility of the group facilitator, and the issue being of high interest to the stakeholders.

Beyond ideal conditions and ideal collaboration between citizens and government representatives, the best way to discover long-term and sustainable working systems is to experiment, since there is no previous experience to tell us if decentralization of a large urban recreation agency can achieve success in the current setting [15]. The present research aims to bring arguments that support the need of future citizen involvement in the decision-making process, especially in establishing and developing parks and green areas within the urban scenery.

## 4. The Role of Parks in Quality of Life Improvement

As the introduction mentioned, in the context of growing urban built areas and decreasing green areas within the city along with an increasing attention granted to urban sustainable development, an emphasis is put on parks and green spaces management as one means of increasing urban habitants' quality of life. For example, Sousa Gomes et al. [16] point out that at the individual level, quality of life is determined by each individual's satisfaction regarding all aspects of their life, among which parks and green spaces were also mentioned. Furthermore, in the noisy and crowded urban setting, parks are gaining more and more popularity as destinations for relaxation [17]. Focusing on parks is also justified by Mani et al. [18], who stated that green areas have a great impact on the way that neighborhoods look and feel for their inhabitants. This issue is even more important as one problem identified in the densified urban areas is the lack of parks and other green areas, as they were among the first venues to be replaced by buildings [2].

Beyond having a major impact on people's health, children's harmonious development and bringing communities together [18], linking parks to the urban sustainable development is also determined by the significant part they occupy in our world: at a global level, parks and green areas account for 12 percent of the Earth's terrestrial surface and 0.5 percent of the Earth's water surface [18]. In addition to their importance in terms of coverage, parks bring all types of advantages to their visitors, just like Eng and Niininen [17] show: the contribution that parks have for their visitors exceeds visible effects, including also a major contribution to the visitors' emotional states which goes beyond the provider's control. On a simpler note, as Beck [19] reveals: our parks, as well as any other public space in our city are an important part of our daily routine and, even though they are open access services, they have a significant effect on our general well-being. Taking into account this perspective, the current research focuses on park visitors' feedback analysis as the first level of involving urban residents in the decision-making process in terms of park management. In order to make it more relevant, the present research focuses on the opinion of park visitors who are also city residents, as their degree of involvement and responsibility towards urban development is higher than of those just passing through the city.

Given the impact parks have on a visitor's life and emotional state, discovering the underlying reasons for visiting a park can reveal parks' growth perspectives in terms of partnership opportunities which can be the basis of providing extra public services, which in turn answer visitors' needs [18]. Even though park visitors usually have no understanding of how a park should be managed and developed, they do experience parks and their services, and can provide useful feedback on how their experience could have been more fulfilling and satisfying [20].

Moreover, parks that are amazing in terms of quality, design and management have a substantial effect on visitors' individual well-being and also create positive outcomes related to the city's image and value [19]. Achieving this aim is even more challenging given that even though there is no denial about parks' impact on health and well-being of all urban living beings, providing enough green spaces that comply with people's needs and wants in terms of green areas is now more challenging than ever when ensuring living and office space is a priority for city developers [2]. Additionally, when taking a strategic approach in park management, perceptions of both frequent and infrequent park visitors

can provide a very useful perspective on how to build a strategic marketing plan that will generate their satisfaction in the long run and their increased willingness to visit more often [17]. The research presented in this paper in the following pages aims precisely at discovering what customers want from the green parks in their city in order to discover how this feedback can become the basis in the decision-making process.

## 5. QoL–Definition, Dimensions, Measurement

Before designing the research methodology, there was a pronounced need to define quality of life (QoL) and its dimensions. Considering the time and space variability related to QoL, we can understand why there is no unanimously accepted definition, nor a classification for its dimensions that can satisfy everyone. Moreover, due to the multidisciplinary feature of this concept pointed out by authors like Raphael [21] and Cummins [22], the definitions address QoL either at society level (the approach of specialists in fields such as sociology, economics or public health services), or at individual level (definitions from psychology, marketing or medicine). Although the first QoL definitions appeared in sociology, the concept grew and has exceeded the limits of this discipline. As Marginean [23] stated in his work about QoL in Romania, limiting the evaluation of this concept to just one discipline's toolkit will be biased.

In his work about quality of life measurement, Constanza [24] gives a definition to QoL from the perspective of the two valences that compose it: objective and subjective. This is the reason for which the author defines this concept as being the degree to which the objective needs of the individual (such as subsistence, security or affection) are satisfied in relation to subjective, individual or group perception of well-being (described in terms of happiness or personal utility). Even if the majority of QoL dimensions are measured in an objective manner, the European Commission recommends also the assessment of the subjective well-being of persons [25]. This double-approach evaluation is based, most often, on several composite indicators, in order to have a strong coverage of what quality of life represents both for the society and the individual [26].

Since the financial crisis started, studies on QoL shifted, first of all, in order to find a better-adapted definition for the concept within the new economic context and, afterwards, find solutions for quality of life improvement. The most important one is the Stiglitz report [27] which led to today's definition of QoL used by the European Commission. This overarching measurement framework simultaneously considers the following 8 + 1 dimensions [28]:

- Material living conditions (income, consumption and material conditions);
- Productive or main activity (quantity and quality of employment and other main activities);
- Physical and/or mental health (health outcome indicators and determinants);
- Education (population's educational completion, skills, participation in life-long learning and opportunities for education);
- Leisure and social interactions;
- Safety—economic and physical (job retention or loss, health problems, aging, global financial and economic crisis, violence and/or crime);
- Governance and basic rights (trust in institutions and public services; discrimination and equal opportunities and active citizenship);
- Natural and living environment (air and water accessibility, noise pollution);
- Overall experience of life (life satisfaction—cognitive appreciation; affect—a person's feelings or emotional states, both positive and negative; and eudaemonic—a sense of having meaning and purpose in one's life, or good psychological functioning).

The last dimension (overall experience of life) represents the subjective approach in this evaluation and it is based on the results from European Quality of Life Survey (EQLS), carried out by Eurofound [29]. This comprehensive study has three major parts [29]: quality of life (how it is perceived by individuals),

quality of public services (perceived in comparison to one's needs) and quality of society (opportunities and threats from the proximity community and society in general).

Another comprehensive analysis of QoL can be found within the OECD Better Life Initiative [30], which features a range of studies and analyses about people's well-being and how to measure it. This interactive Better Life Index evaluates 11 essential areas of material living conditions and QoL: Housing, Income, Jobs, Community, Education, Environment, Civic Engagement, Health, Life Satisfaction, Safety, Work–Life Balance (oecdbetterlifeindex.org).

Apart from these three major studies from Eurostat, Eurofound and OECD, there are also some private institutions' QoL evaluations which have proved their viability throughout time: Economist Intelligence Unit (EIU) with Global Livability Index [31]; Mercer with Quality of Living Index [32] and Monocle with Most Livable City Index [33].

Considering all these academic and practical approaches regarding QoL dimensions, for the present research we have extracted only those indicators or dimensions directly related to park visits, in order to have a better understanding on their relationship and utility for future public policies:

- Health (mental and physical)—one of the major dimensions from all QoL definitions [28–30];
- Social Interaction—indicator used by the European Commission in order to define time spent with other people [28];
- Education and Culture—dimension included in all QoL definitions [28–30];
- Family Life—another indicator used by the European Commission in order to evaluate the social part of one's life [28]; on the other hand, this is one of the indicators that OECD does not take into consideration when assessing QoL, a fact that attracted a series of criticism [34];
- Freedom—indicator of the 'Governance and basic rights' dimension from the European Commission QoL measurement [28], it is also included by Mercer in their Quality of Living Index (within the socio-cultural environment) [32];
- Connection with Nature—indicator within OECD's 'Environment' dimension and European Commission's 'Natural and living environment' dimension [30].

## 6. Park Visitors Survey

### 6.1. Research Methodology

Considering all the above-mentioned associations between quality of life and green spaces, we have developed a quantitative research method in order to identify the degree to which citizens associate park activities with their quality of life. The present research is focusing on the subjective assessment made directly by respondents. The latest Eurofound study about QoL shows how a high proportion of people are reporting difficulties in terms of access to recreational areas in countries like Romania, Portugal or Malta, as it can be seen in Table 1. Comparing Romania's percentage, 29% (highest in EU), with that of Nordic countries (3–4%) motivated us to conduct an applied research focused on Romanian parks.

**Table 1.** People reporting difficulties in accessing neighborhood services (%).

|  | Cultural Facilities | Public Transport | Recycling Facilities | Banking Facilities | Recreational Areas | Groceries |
|---|---|---|---|---|---|---|
| **Highest Proportion** | Romania (55%) Hungary (53%) Portugal (49%) | Finland (36%) France (35%) Portugal (31%) | Romania (41%) Bulgaria (34%) Croatia (30%) | Romania (38%) Hungary (27%) Croatia (24%) | Romania (29%) Portugal (21%) Malta (21%) | Portugal (15%) Czech Republic (15%) Slovenia (14%) |
| **Lowest Proportion** | Netherlands (20%) Cyprus (20%) Denmark (19%) | Netherlands (14%) Slovakia (13%) Luxembourg (11%) | Lithuania (8%) Sweden (7%) Malta (6%) | Cyprus (11%) Germany (10%) Luxembourg (9%) | Finland (4%) Sweden (3%) Denmark (3%) | Slovakia (4%) Bulgaria (4%) Denmark (2%) |

[1] Source: Eurofound, 2016 [29].

We have chosen Herăstrău, Cișmigiu and Carol I parks, the three biggest parks in Bucharest, the country's capital, for two reasons. First of all, these parks are managed by the City Hall, which allows

us to make a series of common recommendations at the end of this paper, recommendations that can lead to an increased collaboration between citizens and government representatives, the best way to discover long-term and sustainable working systems. Second of all, being the biggest parks in Romania, approaching these three parks gives us a broader perspective in terms of visitors' segmentation.

The present research was developed around the main QoL dimensions associated with green spaces and recreational areas—health (mental and physical), social interaction, education and culture, family life, freedom, connection with nature. These dimensions were integrated among the following research objectives:

1. Degree of perceived association between QoL dimensions and park visit;
2. Mental associations made with parks by their visitors (the first thing that comes to their mind when thinking about a specific park);
3. Visiting parks habits (frequency and duration of the visit);
4. Reasons to visit parks;
5. Factors that will lead to an increased frequency of park visits.

As a research method, we have chosen the survey, conducting face-to-face interviews in parks (CAPI-computer assisted personal interview), with a total of 1200 respondents, for a ±3% margin of error and a 95% confidence level. Being a representative survey, the sample was structured based on age, income and gender (using the current data from the National Institute of Statistics [35]), in order to correspond to the area's population structure. Also, we took into consideration the park area in which visitors spend their time, thus the research was structured specifically to cover all the important entrances of Herăstrău, Cișmigiu and Carol I parks.

*6.2. Research Results*

The starting point in this research was represented by the QoL dimensions and their relationship to park visits, in the opinion of people who included this kind of activities in their spare time. As it can be seen in Figure 1 and Table 2, the highest degree of perceived association, on a scale from 1 to 5 (where 5 means very strong), is with mental and physical health—4.72. Although we would have expected the connection with nature would be the strongest, health is the most important QoL dimension overall, which makes people consider it first when deciding on everyday life activities.

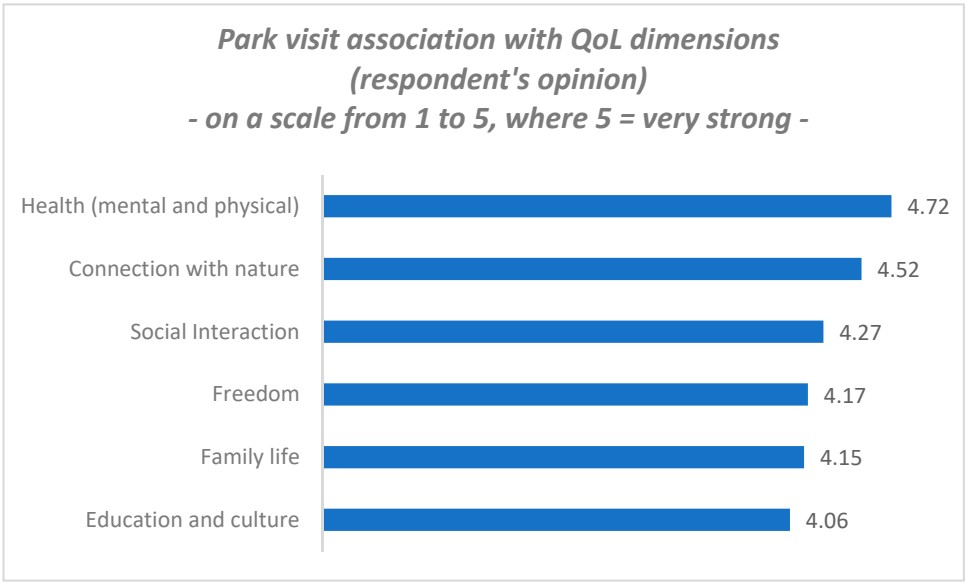

**Figure 1.** Quality of life dimensions associated with park visit (average mean on a scale from 1 to 5).

**Table 2.** Perceived association between QoL dimensions and park visits.

| QoL Dimensions | Mean | Std. Deviation | Coef. of Variation |
|---|---|---|---|
| Health (mental and physical) | 4.72 | 0.57 | 0.1 |
| Connection with Nature | 4.52 | 0.76 | 0.2 |
| Social Interaction | 4.27 | 0.83 | 0.2 |
| Freedom | 4.17 | 0.98 | 0.2 |
| Family Life | 4.15 | 0.96 | 0.2 |
| Education and Culture | 4.06 | 1.01 | 0.2 |

All six dimensions have high coefficients for perceived association with park visits, result which allows us to state that there is a:

1. Very strong association with health and nature;
2. Strong association with social interaction, freedom, family life, as well as education and culture.

These associations are based on the park's image in the consumer's mind, considering that the decision-making process always starts from the already known aspects which influence the way people see and consider products, services, organizations and other options in their life. This is the reason why we asked respondents to tell us the first things that came to their minds when thinking about parks. As can be seen in Table 3, most often people associate parks with nature (23.1%), followed by history/tradition (12.4%) and relaxation (11.6%). Even though health was not directly mentioned by respondents, this can be caused by the fact that health is seen as a result of box-checking the other dimensions, especially if we talk about mental health.

**Table 3.** Mental associations made by park visitors.

| Mental Association | Percent |
|---|---|
| Green/lake/nature | 23.1% |
| History/tradition | 12.4% |
| Relaxation/entertainment/fun | 11.6% |
| Quietness | 7.4% |
| Walking/strolling | 4.6% |

These associations must be, for starters, understood in a general manner, as they represent the average description for the general park visitor. However, there are some differences when we make a segmentation based on park visit frequency and duration. As it can be seen in Tables 4 and 5, before discussing the differences, we have tested the statistical significance of such correlations and in all cases we have less than 0.05 for the *p*-value, which indicates that the differences are statistically significant. Although the Spearman test did not lead to high correlation coefficients, the crosstab shows that, as the frequency and duration of park visit decreases, so does the degree of association between parks and quality of life. This result can be interpreted in two ways: first, the more people visit the park, the better they perceive the association between different aspects of parks and their quality of life; and second, in a reverse logic, the higher the association, the more often people visit parks. Both interpretations can have a major impact on future sustainable public policies, if they are developed within a collaborative setting between citizens and government representatives, as people will see the impact of their decisions in park development, thus leading to an increase in park visit frequency and a stronger perceived association with quality of life.

In order to better understand the individual's approach to park visits, we have asked respondents to describe the reasons for which they visit parks. We can see in Figure 2a (average mean for visits' frequency on a scale from 1 to 5) and Figure 2b (percentage of people) that walking is the most common activity (97.3% of respondents chose it, with a frequency of 4.3), followed by three socially-related activities—hanging out, going to restaurants and spending time with kids. Sport, another health-related QoL activity, alongside walking, is done by $\frac{3}{4}$ of park visitors, being in 5th place in terms of frequency.

**Table 4.** Percent of people perceiving a strong association between QoL dimensions and park visits (crosstab with frequency of park visit).

| QoL Dimensions Associated with Park Visit | Frequency of Park Visit | | | | | | Correlation between Frequency of Park Visit and QoL Dimension | |
|---|---|---|---|---|---|---|---|---|
| | Daily | 2–3 times/week | 1/week | 2–3 times/month | 1/month | Less than 1/month | | |
| Health (Mental and Physical) | 89.1% | 75.4% | 74.5% | 81.6% | 74.1% | 72.7% | Pearson Correlation 0.157 <br> Significance (two-tailed) 0.029 <br> N 1200 | |
| Social Interaction | 61.3% | 48.4% | 51.3% | 50.4% | 33.8% | 35.5% | Pearson Correlation 0.237 <br> Significance (two-tailed) 0.000 <br> N 1200 | |
| Education and Culture | 49.6% | 51.4% | 39.6% | 41.1% | 31.7% | 22.7% | Pearson Correlation 0.261 <br> Significance (two-tailed) 0.000 <br> N 1200 | |
| Family Life | 50.7% | 50.4% | 40.3% | 41.7% | 39.9% | 32.7% | Pearson Correlation 0.215 <br> Significance (two-tailed) 0.049 <br> N 1200 | |
| Freedom | 61.0% | 55.1% | 44.0% | 50.4% | 29.7% | 28.2% | Pearson Correlation 0.366 <br> Significance (two-tailed) 0.049 <br> N 1200 | |
| Connection with Nature | 76.5% | 64.7% | 61.5% | 70.0% | 61.2% | 64.5% | Pearson Correlation 0.119 <br> Significance (two-tailed) 0.043 <br> N 1200 | |

**Table 5.** Percent of people perceiving a strong association between QoL dimensions and park visits (crosstab with visit duration).

| QoL Dimensions Associated with Park Visit | Visit Duration | | | | Correlation between Visit Duration and QoL Dimension | |
|---|---|---|---|---|---|---|
| | More than 3 h | 2–3 h | 1–2 h | Less than 1 h | | |
| **Health (Mental and Physical)** | 84.7% | 81.4% | 62.3% | 86.4% | Pearson Correlation Significance (two-tailed) N | 0.272 0.000 1200 |
| **Social Interaction** | 61.1% | 45.3% | 39.2% | 50.0% | Pearson Correlation Significance (two-tailed) N | 0.248 0.000 1200 |
| **Education and Culture** | 53.2% | 44.3% | 28.8% | 47.7% | Pearson Correlation Significance (two-tailed) N | 0.157 0.049 1200 |
| **Family Life** | 50.6% | 49.8% | 31.0% | 36.4% | Pearson Correlation Significance (two-tailed) N | 0.264 0.000 1200 |
| **Freedom** | 59.6% | 46.8% | 36.4% | 47.7% | Pearson Correlation Significance (two-tailed) N | 0.289 0.000 1200 |
| **Connection with Nature** | 70.2% | 73.3% | 49.0% | 68.2% | Pearson Correlation Significance (two-tailed) N | 0.195 0.000 1200 |

Education and cultural activities, such as reading and going to festivals, have a high percentage in terms of activities done in parks (87.4% and 89.5%, respectively), but a medium score in terms of frequency (3.35 and 3.11). This means that, by including people in the decision-making process of cultural agendas for parks (what type of events should be there, when, how they can be organized, how can visitors participate in the cultural activities, etc.), we can increase the frequency of park visits.

One activity that is directly related to a park's purpose—relaxing on the park's lawns—is at the bottom of the list, both in terms of percentage of people doing it and frequency. From a historical point of view, this situation has deep influences from the communist era, when people were not allowed to step on the grass. As no one told them otherwise since then, people tend to avoid such a potential-conflict type of behavior in parks. The sustainable development of parks should take into consideration this part, as it is directly correlated to one of the main QoL dimension—connection with nature—and involve citizens in an educational campaign about visitors' behavior in parks.

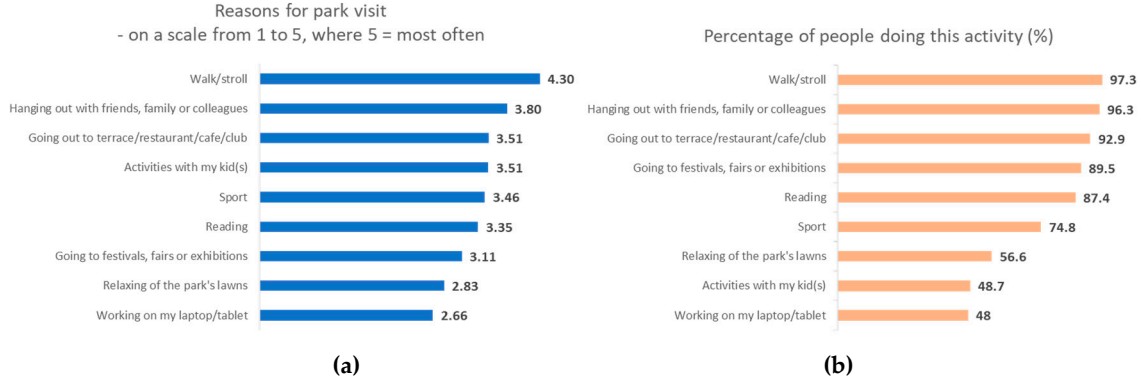

**Figure 2.** (**a**) Reasons to visit parks (frequencies); (**b**) Reasons to visit parks (percentage).

Parks are also important to family life, as we see high frequency for activities related to this—hanging out with family members and doing special activities with kids. Being an important QoL indicator, family life has a major impact on decisions about everyday life activities, thus making parks more welcoming for families should represent a priority for local authorities.

As an open space, the park offers its visitors a higher degree of freedom, where they can express in a personal manner, do things they love and adapt must-do activities, so they become more pleasant, like working on laptops in the park. Although we see that activities related to freedom are not yet so obviously related to parks, this aspect of our life should be exploited in a positive way, giving people the certainty that they can really feel free in parks, in correlation to free access to park's lawns, cultural activities of self-expression and adapting park components to visitors' needs (such as skateparks, off-road tracks, yoga corners, painting and sculpture workshops).

From this analysis we can distinguish an association between activities done in parks and quality of life dimensions, some of them having multiple connections, not just with only one dimension (as will be presented in the final conceptual model of this paper).

All of the above-mentioned recommendations in terms of QoL dimensions and their related activities done by park's visitors are also supported by citizens' proposals when asked what could make them come more often to parks. As seen in Table 6, besides the obvious need for cleanliness, there are also proposals about cultural and sport activities, quietness, as well as family-time-related aspects.

The results obtained from this research led us to the development of a conceptual model which links quality of life, through some of its dimensions, to park visits (analyzed separately through specific activities), as it can be seen in Figure 3.

**Table 6.** Factors that will lead to frequency increase for park visits.

| Factors | Percent |
|---|---|
| If the park was cleaner | 16.5% |
| More concerts/shows/competitions | 16.2% |
| More new activities/attractions/new venues | 10.5% |
| More vegetation | 4.1% |
| More quiet places in the park | 3.9% |
| If my friends and family will meet me here | 3.2% |
| More sports dedicated places | 3.0% |

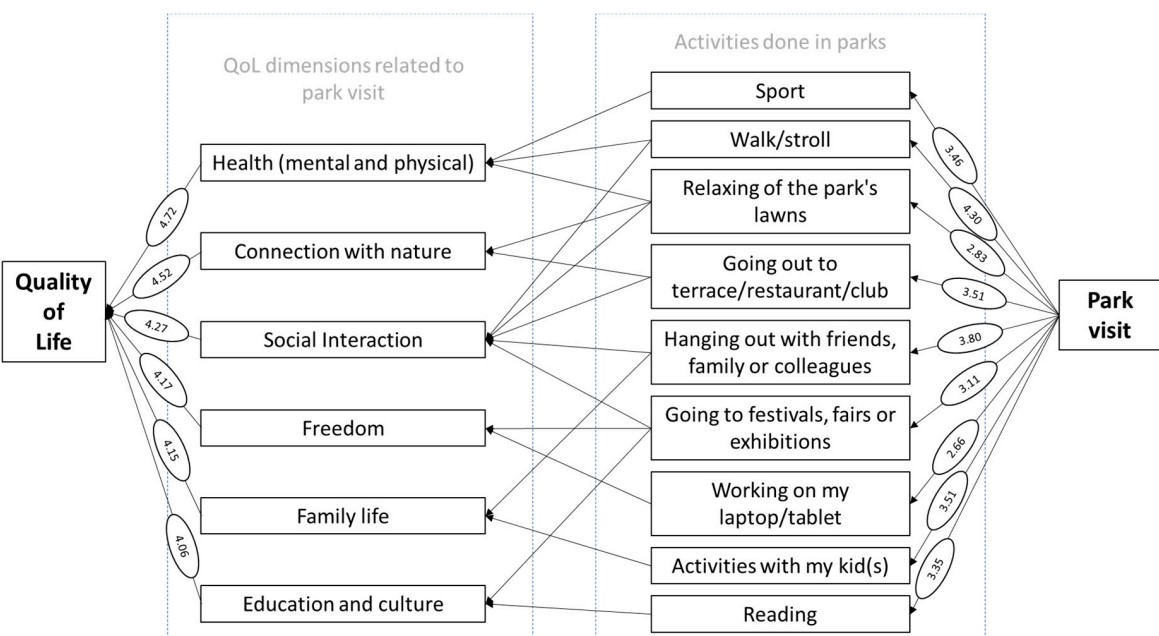

**Figure 3.** Conceptual model on association between park visit and quality of life.

Starting from left to right, one can firstly observe the degree of perceived association of each QoL dimension to park visits, from the highest (health, with 4.72) to the lowest (education and culture, with 4.05). These dimensions are afterwards linked to specific activities done in parks, where the following perceived associations can be found:

- Health (mental and physical)—sport, walk/stroll and relaxing on park's lawns
- Connection with nature—relaxing on park's lawns and going out to terrace/restaurant/club
- Social interaction—walk/stroll, relaxing on park's lawns, going out to terrace/restaurant/club, hanging out with friends, family or colleagues and going to festivals, fairs or exhibitions
- Freedom—going to festivals, fairs or exhibitions and working on my laptop/tablet
- Family life—hanging out with friends, family or colleagues and activities with my kid(s)
- Education and culture—going to festivals, fairs or exhibitions and reading.

In the right part of the model are the average means of each activity's frequency (on a scale from 1 to 5, where 5 means very often). This coefficient, as commented in Figure 2a, indicates the importance of each aspect on the park visit decision-making process, where a collaborative public policy for sustainable development can enhance the potential of each activity for quality of life improvement.

## 7. Public Policy Recommendations

Given our research results and previous experiences in terms of integrating citizens' feedback in the decision-making process, a few recommendations can be mentioned to be taken into consideration for future park and green spaces management in cities similar to Bucharest.

Since a high proportion of citizens choose to visit a park for sport activities or just for walking/strolling and these represent a major component of citizens' health, which is strongly connected to quality of life, major efforts should be directed towards enhancing citizens' active life experience in parks. These might include: maintaining the alleys in good conditions so that walking, skating or biking on them is a pleasant experience, and also creating special spaces dedicated especially to sports-interested visitors (they could have the minimum features in terms of structures, but rather create a welcoming atmosphere for sports-passionate visitors: for example, crossroads water sources and catch-your-breath spots for runners). When we see that respondents defined being in touch with nature as an important component of quality of life, a few recommendations also emerge for enhancing citizens' connection to nature while visiting parks. These include: creating as many visitor friendly lawns as possible, using as many natural materials as possible in building any structures within the park (move to wood and clay rather than plastic and steel), and also paying attention to the whole park experience so that it is as close to real nature as possible (for example, avoid loud noises, or any artificial noise at all, or powerful artificial, human-made smell).

The park also offers the chance to get together with friends and family to its visitors, an important aspect of a balanced life. When focusing on giving the total experience to park visitors in terms of social interaction environment, developing and bringing more events that completely relate and integrate in a green scenery can lead to increased and longer park visits. These events should be separate for different types of customers (for example, festivals or gatherings dedicated to sports enthusiastic people, or for family with toddlers) and should incorporate natural elements (food, appliances and others) as much as possible. Even though these types of events will rarely be organized by local authorities, their responsibility lies in creating the welcoming environment for private sector actors to be interested and willing to bring such events within public parks.

When it comes to funding all these initiatives or recommendations, local authorities have now, more than ever, the need to simplify the citizens' and private sector actors' involvement in park development. Citizens' increased interest in getting involved, as well as private companies' and NGOs' willingness to fund and support sustainable development projects create the perfect premise for opening the doors to collaboration for increased results in terms of park management which in the end result in increased quality of life for the city's population. This trend is also confirmed by Inglis et al. [36] who argue that there are more and more initiatives to develop workable partnerships with the stakeholders as a way to tackle the desire to rely less on government funding.

## 8. Conclusions, Practical Implications and Future Research Directions

Taking into consideration the rapid urbanization and important ecoclimatic challenges, urban sustainable development is more than necessary. Although there is a multitude of definitions for sustainability, it can be also defined in terms of the quality of life and prosperity of citizens. In this article, we argue for increased citizen participation in the sustainability development decision-making process, as it generates better decisions since it implies a wider range of perspectives and also determines people to become better citizens, as they are now partially responsible for the outcome.

Within the urban landscape, well-being depends on urban public spaces, and they should have a coherent development strategy in order to bring value to the community, with economic, social and environmental functions. The research presented in this paper had as a main scope discovering what citizens want from the green parks in their city, related to improving their quality of life, in order to discover how this feedback can be integrated in the decision-making process for urban development.

Considering that there is a multitude of academic and practical approaches regarding quality of life dimensions, we have extracted only those directly related to park visits, in order to have a better understanding of their relationship and utility for future public policies—health (mental and physical), social interaction, education and culture, family life, freedom, and connection with nature.

Our research showed that all of these six dimensions have high coefficients for perceived association with park visits, a fact that allows us to make the interpretation of a very strong association with health

and nature; and a strong association with social interaction, freedom, family life, as well as education and culture.

One of the most important research outcomes is that, as the frequency and duration of park visit decreases, so does the power of association between parks and quality of life. This fact can be interpreted in two ways: first, the more people visit the park, the better they perceive the association with their quality of life; and the second, in a reverse logic, the more people see the association, the more often they visit parks. Both interpretations can have a major impact in future sustainable public policies, if they are developed in a collaborative way, between citizens and government representatives, as people will see the impact of their decisions in park development, thus leading to increase in park visit frequency and a stronger perceived association with quality of life.

The results obtained from this research led us to the development of a conceptual model that links quality of life, through its six dimensions, to park visits (analyzed separately through specific activities). This model can and should be used by Bucharest public authorities (such as City Hall and park administration) in order to build a collaborative process for urban sustainable development, where actions can be taken in terms of enhancing citizens' active life experience in parks, their connection to nature while visiting parks, as well as developing and bringing more events to the park, that completely relate and integrate in a green scenery.

When considering the limits of the present research, we can mention at least two which are important for a higher level of relevance. First of all, the fact that we conducted face-to-face interviews with visitors from Bucharest (the country's capital city) parks only, provides us with information about the top tier, leaving out the rest of the country, where most probably the situation regarding green areas is worse. This limit also relates to the fact that the specific national context will prevent us from extrapolating the result to other countries. The second limit refers to the fact that evaluating people's perceptions about park visit's impact on quality of life leads to a subjective assessment, which can be sometimes far from the real situation.

Considering these limits, the main recommendation for future research is to build a before and after experiment, in which people's QoL is evaluated at the start of the research, in order to compare the same indicators after a determined period of time of exposure to parks. This way QoL impact is determined in an objective manner. Another recommendation for future research will be to actually test the correlations for statistical significance, when respondents are asked to evaluate their current QoL level, not only to make a subjective assessment about the association between QoL and park visits.

**Author Contributions:** Conceptualization, M.C. and Ș.-C.C.; data curation, M.C. and A.P.; formal analysis, M.C.; funding acquisition, Ș.-C.C.; Investigation, M.C. and Ș.-C.C.; methodology, M.C. and Ș.-C.C.; project administration, M.C.; supervision, M.C. and Ș.-C.C.; validation, M.C. and Ș.-C.C.; visualization, M.C. and A.O.; writing—original draft, M.C., A.O., Ș.-C.C. and A.P.; writing—review and editing, M.C., A.O. and A.P.

**Funding:** This work was supported by a grant of the Romanian Ministry of Research and Innovation, UEFISCDI, project number PN-III-P1-1.2-PCCDI-2017-0800/86PCCDI/2018-FutureWeb, within PNCDI III. The APC was funded also by project PN-III-P1-1.2-PCCDI-2017-0800/86PCCDI/2018.

**Conflicts of Interest:** The authors declare no conflict of interest. The funders had no role in the design of the study; in the collection, analyses, or interpretation of data; in the writing of the manuscript, or in the decision to publish the results.

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
