# Peer review of "Sustainable Development of Urban Green Areas for Quality of Life Improvement—Argument for Increased Citizen Participation"

_sustainability, doi:10.3390/su11184868_

Round 1

Reviewer 1 Report

The authors investigated the perceived association between part visit and quality of life in 1200 subjects, and by asking what subjects want from the green parks, aimed to discover how feedback from citizens can be integrated in the decision-making process for urban sustainable development. The approach of theoretical reasoning combined with field investigation (face to face interviews) is the main strength of the paper. However, I have three major concerns and a few minors issues to be addressed by the authors.

Major concerns:

1, The description of the "strong correlation" is misleading and confusing, it is not statistical analysis but correlation in people's subjective report, right? By "quality of life dimensions associated with park visit", do the authors mean QOL level due to park visits, or just the subjective belief that QOL is associated with park visit? this should be stated clearly. Furthermore, did the authors investigate the QOL per se with each subject? I mean statistically speaking, QOL per se is a more important variable, whose association with park visit frequencies and durations is subject to objective, statistical testing. Right now I'm not sure what the authors actually measured and consequently, I'm not sure what statistical tests they should conduct with this variable.

2, I think the purpose of the authors showing Table 3 and 4 is to look the association between "QOL dimensions associated with park visit" (as I said above, I'm not sure what this means but whatever it means, I'll have this concern) and actual park visits. If this is true, then I would suggest the authors conduct a statistical correlation analysis between "QOL dimensions associated with park visit" and frequency of park visit as well as visit duration to detect any subjective relations, to support the authors' arguments. More inference statistics rather than descriptive statistics are required.

3, The same comment of 2 applies to Figure 4, the conceptual model on association between park visit and QOL. The authors added a lot links in the middle of the model, but these should be (better) tested with data. The authors already got data on both the left and the right and what they left to do is to conduct a statistical correlation.

Minor:

1, As already stated in my first major concern, how the authors framed the questions is less clear to me, for example, "mental associations made with parks" (what is a mental association? can you put the original question here?), "reasons to visit parks" (reasons of past visits or the current visit?), "percentage of people doing this activity" reported in Figure 3 (during the current visit or when?).

2, Basically, descriptive data of the mean should be presented as mean ± SD.

3, Page 6 line 269, what is "for a +/-3% margin of error and a 95% confidence level"?

4, The left part of Figure 1 is partially missing.

5, It seems to me that in its current form, the title of the paper does not comprehensively represent the whole content especially the data presented in this paper.

Author Response

The description of the "strong correlation" is misleading and confusing, it is not statistical analysis but correlation in people's subjective report, right?

By "quality of life dimensions associated with park visit", do the authors mean QOL level due to park visits, or just the subjective belief that QOL is associated with park visit? this should be stated clearly. 

Furthermore, did the authors investigate the QOL per se with each subject? I mean statistically speaking, QOL per se is a more important variable, whose association with park visit frequencies and durations is subject to objective, statistical testing. Right now I'm not sure what the authors actually measured and consequently, I'm not sure what statistical tests they should conduct with this variable.

The reviewer is right, as we haven’t conducted statistical tests in order to measure the degree of correlation between these two variables, but rather presented the perceived association in the mind of the park visitor.

Thus, in the revised version of our article we have replaced ‘correlation’ with ‘perceived association’.

In this research, when speaking about ‘quality of life dimensions associated with park visit’, we mean QoL dimensions that visitors feel that are influenced somehow by park visits; it’s a subjective assessment made directly by respondents.

We have included this information in the first paragraph of 6.1. Research methodology.

The questionnaire is not measuring directly the QoL, as such a measurement will require more than a simple question about how people are perceiving their QoL level. Having done a lot of QoL researches in the past, the leading co-author Mihaela Constantinescu decided to skip self-evaluation of one’s QoL. The first reason was that QoL can be measured only by assessing every aspect of one’s life, not just by asking them to rate their QoL level at the moment. The second reason for which QoL level wasn’t a variable in the present research relates to the fact that, in order to identify the objective correlation between park visits and QoL level, we should conduct a before and after experiment, in order to truly see the differences, not just ask people to tell us if their QoL is better or worse due to park visits. Having all this in mind, the authors focused for now on which QoL dimensions do respondents fell that are connected with visiting parks from their city of residence.

I think the purpose of the authors showing Table 3 and 4 is to look the association between "QOL dimensions associated with park visit" (as I said above, I'm not sure what this means but whatever it means, I'll have this concern) and actual park visits. If this is true, then I would suggest the authors conduct a statistical correlation analysis between "QOL dimensions associated with park visit" and frequency of park visit as well as visit duration to detect any subjective relations, to support the authors' arguments. More inference statistics rather than descriptive statistics are required.

In order to make the correlations from Table 3 and 4 more correct, we have now included also the results of the Chi-square test. Also, in the text corresponding to these 2 tables we have included the following addition:

‘As it can be seen in Table 3 and Table 4, before discussing the differences, we have tested the statistical significance of such a correlation and in all cases we have less than .05 for the Chi-square coefficient, which indicates that the differences are statistically significant.’

The same comment of 2 applies to Figure 4, the conceptual model on association between park visit and QOL. The authors added a lot links in the middle of the model, but these should be (better) tested with data. The authors already got data on both the left and the right and what they left to do is to conduct a statistical correlation.

Considering that all the coefficients from Figure 4 are actually not correlation coefficients, but rather average mean for the perceived association made by respondents between QoL dimensions and park visits, we haven’t conducted any test for correlation statistical significance. We are aware of the fact that this is a limit of our research, and we have mentioned this in section 8. Conclusions, practical implications and future research directions:

‘When considering the limits of the present research, we can mention at least two which are important for a higher level of relevance. First of all, the fact that we conducted face to face interviews with visitors from Bucharest (country’s capital city) parks only, provides us with information about the top tier, leaving out the rest of the country, where most probably the situation about green areas is worse. This limit also relates to the fact that the specific national context will prevent us from extrapolating the result to other countries. The second limit refers to the fact that evaluating people’s perceptions about park visit’s impact on quality of life leads to a subjective assessment, which can be sometimes far from the real situation.

Considering these limits, the main recommendation about future researches is to build a before and after experiment, in which people’s QoL is evaluated at the start of the research, in order to compare the same indicators after a determined period of time of exposure to parks. This way QoL impact is determined in an objective manner. Another recommendation for future researches will be to actually test the correlations for statistical significance, when respondents are asked to evaluate their current QoL level, not only to make a subjective assessment about the association between QoL and park visits.’

As already stated in my first major concern, how the authors framed the questions is less clear to me, for example, "mental associations made with parks" (what is a mental association? can you put the original question here?), "reasons to visit parks" (reasons of past visits or the current visit?), "percentage of people doing this activity" reported in Figure 3 (during the current visit or when?).

-        "mental associations made with parks" (what is a mental association? can you put the original question here?)

Question used in the survey: ‘Which is the first thing that comes to your mind when thinking about this park?’

With this question we intended to identify top of mind associations.

-        "reasons to visit parks" (reasons of past visits or the current visit?)

This was a question that targeted respondent’s usual behavior in the park, so we asked why he usually comes to that park.

-        "percentage of people doing this activity" reported in Figure 3 (during the current visit or when?).

In the same situation as above, as we have asked respondents to share with us what the usually do in that park.

Basically, descriptive data of the mean should be presented as mean ± SD.

We have included this information now in the descriptions of tables and figures. Also, we have included table 2 which presents the mean, along side the std. deviation and coef. of variation.

Page 6 line 269, what is "for a +/-3% margin of error and a 95% confidence level"?

These are the statistical conditions established in order to determine sample size.

The left part of Figure 1 is partially missing.

We have corrected this error.

It seems to me that in its current form, the title of the paper does not comprehensively represent the whole content especially the data presented in this paper.

Hopefully, now that we have made all these major revisions to our paper, based on both reviews received, the paper is correctly presenting the data and responding to its title.

Reviewer 2 Report

-Line 31: do you have literature showing that perceived crowding is a major impetus for sustainable urban development research? 

-Line 36: “discovering these” what? 

-Line 37: there is extensive literature defining interdisciplinary and transdisciplinary research. Even if you are only using the Wagner material to support this statement, you should explain what you mean by these terms. They are not mentioned again throughout the article so they kind of hang in the introduction without much meaning. 

-Line 45: At this point in the paper, the phrase “more and more” has been used five times already. Please consider exploring other adjectives to describe growth

-Line 51: At this point, the goals of sustainable urban development have been touched upon, however, the authors should be direct in their introduction and in the body of the article about what definition or aspect of urban sustainable development they are focusing on. 

-LIne 71: awkward phrasing: " citizens’ involvement in the decision-making process is so much needed”

-Line 99: Research doesn’t aim to prove 

-The end of 121-126: these lines should be rewritten and combined. As they read, they seem like disparate examples yet they are referring to similar challenges. 

-Line 140: grammar issues 

-line 154: grammar issue 

-Line 154-176: It is not clear whether the aim is to outline park benefits for urban residents or visitors throughout this whole paper. If the premise of the paper is to argue for more citizen participation in sustainable urban development processes, it seems that you are focusing on urban residents. There needs to be more distinction between the stakeholders being examined in this paper. 

-Line 206: why not just say 9 dimensions?

-Line 231-232: examples of colloquial writing throughout the paper 

-Line 233: grammar issue at the beginning of the line 

-line 235-238: even though you extracted these dimensions, you still need to identify which sources these ultimately came from.

-Line 241: Up until this point, it is still unclear if you are focusing on visitors or residents who use parks. If you focused on residents (which one would assume since you are interested in citizens’ collaboration with local government), then how did you distinguish them from non-residents in the park. 

-Line 268: but how did you approach people? For example, was it an exit survey? 

-Line 270: what statistics did you use to compare your sample to in order to confirm representativeness? 

-Line 276-277: why did you expect that connection with nature would be the strongest perceived association? Is there previous research supporting this? it seems like this is something better placed in the discussion section 

-Line 283: grammar 

-Where are the rest of the labels on the y-axis in Future 1 

-Line 285: why is the health and nature dimension also italicized? 

-Line 321: need to be consistent throughout the paper about whether you are spelling out numbers or not 

-Line 338: grammar 

-Table 5 shows inconsistency in the formatting of tables 

-The results do not reflect a correlation analysis, rather they reflect a frequency analysis. Correlation analysis requires reporting r, the significance of relationships, descriptive statistics for each variable, and the R square

-Line 377: it does not appear that correlation coefficients are being reported, rather frequency results seem to be represented 

-Line 403: visitors are described as customers? 

-Line 4121-422: grammar 

-Why weren’t visitors asked about their perceived role of local government in park maintenance and access? Or, why weren’t visitors asked about their perceived empowerment in government decisions about parks and green spaces? 

Author Response

-Line 31: do you have literature showing that perceived crowding is a major impetus for sustainable urban development research? 

We have added references to the original phrase:

‘Sustainable urban development has been a hot topic in the recent years since more and more people are concerned with their quality of life within the crowded urban setting.’

Resulting in this phrase:

‘Sustainable urban development (as well as social sustainable development [1]) has been a hot topic in the recent years since more and more people are concerned with their quality of life within the crowded urban setting [2]’

-Line 36: “discovering these” what? 

‘These’ refer to the answers to these 2 questions:

-        What does sustainable development mean when it comes to parks?

-        What does citizens’ participation imply in transforming park across cities?

-Line 37: there is extensive literature defining interdisciplinary and transdisciplinary research. Even if you are only using the Wagner material to support this statement, you should explain what you mean by these terms. They are not mentioned again throughout the article so they kind of hang in the introduction without much meaning. 

To the original phrase:

‘For discovering these, science and citizens’ feedback collected through research based on interdisciplinary and transdisciplinary transformation can lead to more realistic and actionable results [1]. Taking this last argument into consideration’

We have added the following:

‘For discovering these, science and citizens’ feedback collected through research based on interdisciplinary and transdisciplinary transformation can lead to more realistic and actionable results [1]. While interdisciplinary transformations imply bringing together different disciplines researchers’ action and integration, multidisciplinary transformations extend contributions to combine knowledge and skills of academics and other stakeholder from the civic society [2]. Both of these approaches might bring a consistent contribution that will lead the way on the path towards creativity in urban sustainable development. Taking this last argument into consideration’

-Line 45: At this point in the paper, the phrase “more and more” has been used five times already. Please consider exploring other adjectives to describe growth

We have replaced the following:

-        ‘raises more and more awareness’ (line 32) with ‘raise increased awareness’

-        ‘people choosing nowadays more and more urban areas’ (line 41) with ‘People progressively choosing nowadays urban areas’

-        ‘as people are moving more and more towards urban settings’ (line 45) with ‘as people are moving towards urban settings’

-Line 51: At this point, the goals of sustainable urban development have been touched upon, however, the authors should be direct in their introduction and in the body of the article about what definition or aspect of urban sustainable development they are focusing on. 

We have added to that last phrase of the last paragraph:

‘Basically, caring for all the beings in the urban setting is justified by their belonging, along with humans, to the entire urban ecosystem [4].’

The following:

‘In the current research framework, urban sustainable development is defined as ‘positive quantitative, qualitative, directed, irreversible changes in the process of supply-production-distribution-consumption, the ones that help to adapt to the effects of endogenous and exogenous factors, to ensure higher rates of reproduction of resources relatively to the rate of consumption and, at the same time, to eliminate the disturbance of socio-ecological and economic security regarded as a result of the balance between consumption and reproduction of resources’ [5]’

-Line 71: awkward phrasing: " citizens’ involvement in the decision-making process is so much needed”

We have rephrased that paragraph as it follows:

‘an ideal solution would need to combine authentic, people-oriented public policies and opening the door to citizens’ involvement in the decision-making process’

instead of

‘combining authentic, people-oriented public policies with opening the door to citizens’ involvement in the decision-making process is so much needed’

-Line 99: Research doesn’t aim to prove 

We have rephrased ‘confirm this belief proving that citizen participation’ to this version: ‘confirm this assumption, discovering that citizen participation’

-The end of 121-126: these lines should be rewritten and combined. As they read, they seem like disparate examples yet they are referring to similar challenges. 

We have rephrased:

‘When it comes to practical experiments with citizens’ participation, experiences of South-Eastern European countries show insufficient cooperation among politicians, citizens and local administration to establish a legal framework for achieving higher levels of active participation in the policy-making process [11]. Additionally, Godbey & Kraus [12] show that when it comes to real life experience, citizens’ involvement in decision-making process has been rather shallow.’

To this version:

‘When it comes to practical experiments with citizens’ participation, Godbey & Kraus [12] point out that citizens’ involvement in decision-making process has been rather shallow, a conclusion also confirmed in the South-Eastern European countries case where there is insufficient cooperation among politicians, citizens and local administration to establish a legal framework for achieving higher levels of active participation in the policy-making process [11]’

-Line 140: grammar issues 

We have rephrased:

‘The present research aims to bring arguments for the need of further experiments involving citizen in decision-making process’

To this version:

‘The present research aims to bring arguments that support the need of future citizen involvement in the decision-making process’

-line 154: grammar issue 

We have corrected to ‘children’s harmonious development’

-Line 154-176: It is not clear whether the aim is to outline park benefits for urban residents or visitors throughout this whole paper. If the premise of the paper is to argue for more citizen participation in sustainable urban development processes, it seems that you are focusing on urban residents. There needs to be more distinction between the stakeholders being examined in this paper. 

We have added to that last phrase of the last paragraph:

‘Even though park visitors usually have no understanding of how a park should be managed and developed, they do experience parks and their services and can provide a useful feedback on how their experience could have been even more fulfilling and satisfying [16].’

The following:

‘Taking into account this perspective, the current research focuses on park visitors’ feedback analysis as the first level of involving urban residents in the decision-making process in terms of park management. In order to make it more relevant, the present research is focusing on the opinion of park visitors that are also city residents, as their degree of involvement and responsibility towards urban development is higher than of those just passing through the city.’

-Line 206: why not just say 9 dimensions?

This ‘8+1’ classification for quality of life dimensions is proposed by the European Committee, as the latter one is more of a effect of the previous eight ones, where the individual’s subjective level of satisfaction with quality of life is measured.

-Line 231-232: examples of colloquial writing throughout the paper 

This problem was resolved with the English language editing.

-Line 233: grammar issue at the beginning of the line 

EU NU VAD LA CE SE REFERA (si m-am uitat in articolul original evaluat de ei, ca sa nu gresesc nr liniei)

-line 235-238: even though you extracted these dimensions, you still need to identify which sources these ultimately came from.

We have rephrased:

‘Considering all these academic and practical approaches regarding QoL dimensions, we have extracted only those dimensions directly related to park visits, in order to have a better understanding on their correlation and utility for future public policies – Health (mental and physical), Social Interaction, Education and culture, Family life, Freedom, Connection with nature’

To this version:

‘Considering all these academic and practical approaches regarding QoL dimensions, for the present research we have extracted only those indicators or dimensions directly related to park visits, in order to have a better understanding on their correlation and utility for future public policies:

-        Health (mental and physical) – one of the major dimensions from all QoL definitions [28][29][30];

-        Social Interaction – indicator used by the European Commission in order to define time spent with other people;

-        Education and culture – dimension included in all QoL definitions [28][29][30];

-        Family life – another indicator used by the European Commission in order to evaluate the social part of one’s life; on the other hand, this is one of the indicators that OECD doesn’t take into consideration when assessing QoL, fact that attracted a series of criticism [31];

-        Freedom – indicator of ‘Governance and basic rights’ dimension from the European Commission QoL measurement, it is also included by Mercer in their Quality of Living Index (within the socio-cultural environment);

-        Connection with nature – indicator within OECD’s ‘Environment’ dimension and European Commission’s ‘Natural and living environment’ one.’

-Line 241: Up until this point, it is still unclear if you are focusing on visitors or residents who use parks. If you focused on residents (which one would assume since you are interested in citizens’ collaboration with local government), then how did you distinguish them from non-residents in the park. 

We have addressed this problem by including the following sentence earlier in the text (when responding to problem from Line 154-176): In order to make it more relevant, the present research is focusing on the opinion of park visitors that are also city residents, as their degree of involvement and responsibility towards urban development is higher than of those just passing through the city.

In order to make sure that all respondents were city residents, we had a filter about this and also asked them which in the neighborhood in which they live.

-Line 268: but how did you approach people? For example, was it an exit survey? 

Respondents were approached when entering the park by one of the operators holding a tablet in his hand, as we used a CAPI - computer assisted personal interview. In the article we now included an extra information in order to explain also how people were approached:

‘Also, we took into consideration the park area in which visitors spend their time, thus the research was structured so it will cover all the important entrances for Herăstrău, Cișmigiu and Carol I parks.’

-Line 270: what statistics did you use to compare your sample to in order to confirm representativeness? 

We have rephrased:

‘Being a representative survey, the sample was structured based on age, income and gender, in order to correspond to the area’s population structure.

To this version:

‘Being a representative survey, the sample was structured based on age, income and gender (using the current data from the National Institute of Statistics [32]), in order to correspond to the area’s population structure.’

-Line 276-277: why did you expect that connection with nature would be the strongest perceived association? Is there previous research supporting this? it seems like this is something better placed in the discussion section 

This expectation arises from the fact that park visits are included in the QoL dimension of ‘Natural and living environment’, thus being the closest one to quality of life.

-Line 283: grammar 

This problem was resolved with the English language editing.

-Where are the rest of the labels on the y-axis in Future 1 

We have resolved this issue by replacing the figure with the correct one.

-Line 285: why is the health and nature dimension also italicized? 

We have resolved this issue.

-Line 321: need to be consistent throughout the paper about whether you are spelling out numbers or not 

We have replaced ‘fifth’ with ‘5th’.

-Line 338: grammar 

We have resolved this issue.

-Table 5 shows inconsistency in the formatting of tables 

We have resolved this issue.

-The results do not reflect a correlation analysis, rather they reflect a frequency analysis. Correlation analysis requires reporting r, the significance of relationships, descriptive statistics for each variable, and the R square

You are right. The results reflect visitor’s perception on the correlation (association) between park visits and different QoL dimensions.

In order to make this fact clearer, we have replaced ‘correlation’ with ‘perceived correlations’, in order to highlight the fact that there isn’t a statistical correlation identified by the authors, but rather a correlation in the mind of the respondent.

-Line 377: it does not appear that correlation coefficients are being reported, rather frequency results seem to be represented 

You are right. In order to make the results clearer, we have rephrased:

‘In the right part of the model there are the coefficients derived from each activity’s frequency (on a scale from 1 to 5, where 5 means very often).’

To this version:

‘In the right part of the model are the average means of each activity’s frequency (on a scale from 1 to 5, where 5 means very often).’

-Line 403: visitors are described as customers? 

In terms of marketing, the park visitor can be seen as a customer, if we think that he/she can benefit from products and services offered in the park area (like food, drinks, shows, etc.).

-Line 4121-422: grammar 

This problem was resolved with the English language editing.

-Why weren’t visitors asked about their perceived role of local government in park maintenance and access? Or, why weren’t visitors asked about their perceived empowerment in government decisions about parks and green spaces? 

This is a very good point and we will continue the present research with a one focused only of citizen involvement in government decisions, as in Romania this is a very important issue which requires a detailed discussion, in order to overcome the limits of socially desired answers.

Round 2

Reviewer 1 Report

Thank the authors for making the changes. They have now addressed most of my concerns and although the perceived association is not statistical correlations, the results may provide helpful insights for future research.

One last but not still important issue, the authors interpreted the results of the chi-square analysis in Table 4 and Table 5 as if they have conducted a correlation analysis, which should be corrected. In this case, correlation analysis rather than chi-square is more appropriate for their purpose. Meanwhile, the authors should report all the important statistics rather than p value alone.

Author Response

One last but not still important issue, the authors interpreted the results of the chi-square analysis in Table 4 and Table 5 as if they have conducted a correlation analysis, which should be corrected. In this case, correlation analysis rather than chi-square is more appropriate for their purpose. Meanwhile, the authors should report all the important statistics rather than p value alone.

We have replaced the previous chi-square test with a bivariate correlation analysis (Spearman), including in Tables 4 and 5 all the data: Pearson correlation, Sig. (2-tailed), N.

We also made a slide change in the text referring to these two tables:

‘As it can be seen in Table 3 4 and Table 45, before discussing the differences, we have tested the statistical significance of such a correlation and in all cases we have less than .05 for the p coefficient, which indicates that the differences are statistically significant. Although the Spearman test didn’t lead to high correlation coefficients, the crosstab shows that, as the frequency and duration of park visit decreases, so is the degree of association between parks and quality of life.’

Reviewer 2 Report

-Make sure to conduct a spell check before final submission. For instance, Line 16, “associated” is still misspelled in red. Line 39, “simply” is misspelled in red.   -Be consistent with whether you are using “urban sustainable development” or “sustainable urban development”   -Line 38-43 is an incorrect summary of their cited article and the general conceptualization of multi-, inter, and trans-disciplinary research.    -Line 46, please use data to support this claim. Particularly, what are the urban migration statistics in Romania?    -Line 75-79, there is an automatic jump to climate change issues. It seems out of place and not well contextualized    -Line 196, be consistent in whether you are capitalizing all words in a heading or not    -Line 202-203, How are specialists in fields such as sociology, economic, or public health services representative of societal view of quality of life? I am also not sure what is meant by “individual level” views.    -Line 254, still need a citation for this QOL dimension. In general, even though you mention the OECD and the European Commission, you need citations for their definitions.    -Line 312 and 342, make sure to insert page breaks so that tables are not cut across pages 

Author Response

-Make sure to conduct a spell check before final submission. For instance, Line 16, “associated” is still misspelled in red. Line 39, “simply” is misspelled in red.  

We have made the requested corrections and have conducted an extensive spell check

-Be consistent with whether you are using “urban sustainable development” or “sustainable urban development”

We have replaced all references to ‘sustainable urban development’ with ‘urban sustainable development’

-Line 38-43 is an incorrect summary of their cited article and the general conceptualization of multi-, inter, and trans-disciplinary research.   

We have replaced the initial paragraph:

‘While interdisciplinary transformations imply bringing together different disciplines researchers’ action and integration, multidisciplinary transformations extend contributions to combine knowledge and skills of academics and other stakeholder from the civic society [4].’

With the following one:

‘While interdisciplinary studies involve researchers in different disciplines accepting concerted action and integration in order to achieve a shared goal related to a common study subject, transdisciplinary contributions combine knowledge and concepts used by academics and researchers, but also by other stakeholders from the civic society [4].’

-Line 46, please use data to support this claim. Particularly, what are the urban migration statistics in Romania?   

We have added a few details about this to the following paragraph:

‘People progressively choosing nowadays urban areas as their home led to an increasing urban population’

Resulting in:

‘People progressively choosing nowadays urban areas as their home led to an increasing urban population (for example, in Romania according to the National Institute of Statistics, the migration rate from rural to urban setting has been steadily growing between 2011 and 2018, hitting a level of 7.2 per 1000 inhabitants, very close to the 7.5 level of 2010).’

-Line 75-79, there is an automatic jump to climate change issues. It seems out of place and not well contextualized   

We have rephrased the following paragraph:

‘Therefore, in order to achieve sustainability objectives, city managers and stakeholders must undertake steps to deal with climate change issues (upon which cities have a specific and significant role in terms of impact) and the effects it has on the entire population and environment, while keeping in mind the need to build a complete trust-based ecosystem between locals and the environment [6].’

To this version:

‘Therefore, in order to achieve sustainability objectives, city managers and stakeholders must focus on the need to build a complete trust-based ecosystem between locals and the environment [6].’

-Line 196, be consistent in whether you are capitalizing all words in a heading or not 

We have modified the heading, capitalizing all words.

-Line 202-203, How are specialists in fields such as sociology, economic, or public health services representative of societal view of quality of life? I am also not sure what is meant by “individual level” views. 

This distinction in QoL definitions between society (we have replaced ‘societal’ with ‘society’) and individual level refers to the manner in which authors see the quality of life – either a collective item or an individual one. Sociology, economic and public health services specialists use most of the time indicators that define the quality of life of the society/community/region/country, whereas specialists from domains such as psychology, marketing or medicine tend to focus on the individual and his QoL (through indicators such as personal health, happiness, interests and fulfillments).

-Line 254, still need a citation for this QOL dimension. In general, even though you mention the OECD and the European Commission, you need citations for their definitions.   

We have resolved this problem, by presenting the citation for each information included in this paragraph.

-Line 312 and 342, make sure to insert page breaks so that tables are not cut across pages 

We will see that the final version of our article has the best shape and structure. After presenting the particle with track changes for each reviewer, we will make the final presentation for publication.